# Efficacy Testing of a Multi-Access Metaverse-Based Early Onset Schizophrenia Nursing Simulation Program: A Quasi-Experimental Study

**DOI:** 10.3390/ijerph20010449

**Published:** 2022-12-27

**Authors:** Sun-Yi Yang, Mi-Kyung Kang

**Affiliations:** 1College of Nursing, Medical Campus, Konyang University, 158, Gwanjeodong-ro, Seo-gu, Daejeon 35365, Republic of Korea; 2College of Nursing, Chungwoon University, 25 Daehak-gil, Hongseong-eup, Hongseong-gun 32244, Chungnam, Republic of Korea

**Keywords:** nursing education, user computer interface, patient simulation, psychiatric nursing, nursing–patient relations

## Abstract

This study investigated the effects of a multi-access, metaverse-based early onset schizophrenia nursing simulation program based on Raskin and Rogers’ person-centered therapy. The program’s effectiveness was tested using a nonequivalent control group pre-test/post-test design. A quasi-experimental simulation study with both pre- and post-test designs was adopted. The experimental group (*n* = 29) used the simulation program, whereas the control group (*n* = 29) received only an online lecture on schizophrenia nursing. Changes in scores among experimental and control groups were compared using independent *t*-tests and analyses of covariance with PASW SPSS-WIN 27.0. Post-intervention, the knowledge regarding patients with early onset schizophrenia, critical thinking ability, and the ability to facilitate communication increased significantly in the experimental group compared with the control group. The nursing simulation program for children with early onset schizophrenia using a metaverse improved nursing students’ knowledge, critical thinking ability, and ability to facilitate communication. This training method should be adapted without spatiotemporal constraints by partially supplementing clinical and simulation-based practice. In clinical nursing training, metaverse technical limitations should be identified, and training topics should be selected. Employing EduTech in a metaverse environment can provide clinical education to nurses in psychiatric wards and improve therapeutic communication with their psychiatric patients.

## 1. Introduction

Coronavirus disease 2019 (COVID-19) has swept the world ahead of the fourth industrial revolution and accelerated the transition to a new educational paradigm of information technology (IT)-based e-learning in nursing education [1]. Various attempts have been made to develop alternative nursing practice education programs to replace nursing students’ clinical practice [2]. However, clinical practice in psychiatric units is unlikely to resume in South Korea given psychiatric patients’ vulnerability to COVID-19 infections owing to a typical lack of self-hygiene and closed psychiatric wards [2].

As psychiatric nursing education aims to develop the ability to provide nursing care to people with compromised mental health, nursing students must develop the ability to build a rapport with patients and engage in therapeutic communication [3]. Among psychiatric patients, those with schizophrenia are characterized by both positive symptoms, such as hallucinations, delusions, disorganized speech, and abnormal behaviors; and negative symptoms, such as a lack of social skills, apathy, and a blunt affect [4]. Farahat Abd Alaziz et al. [4] demonstrated that these aspects of patients with schizophrenia instill fear and anxiety in nursing students, often hampering their efforts to build rapport and communicate with these patients. Similarly, patients with schizophrenia can experience anxiety during consultations and communications with nursing students [4]. Therefore, efficient therapeutic communication has a significant, positive effect on controlling negative thoughts in patients with schizophrenia, while failures in communication cause failures in treatment [5].

The person-centered therapy proposed by Raskin and Rogers [6] is defined as a horizontal relationship between the counselor and the client, and the counselor empathizing and showing positive interest and acceptance of the client to form a therapeutic relationship. Three key concepts have been proposed: sincerity, unconditional positive respect, and empathy, which are based on the relationship between the therapist and the client, as therapeutic change occurs only in the context of the relationship when the therapist treats and empathizes with the client truthfully. However, by deeply empathizing with clients, they should be able to perceive and experience their own potential and make meaningful and lasting changes [7]. Communication with a patient with early onset schizophrenia must be therapeutic and incorporate these three core conditions. Among schizophrenia types, schizophrenia with onset before adulthood is also called “early onset” schizophrenia. It is often preceded by nonspecific psychosocial disorders, and patients display weak positive symptoms [8]. Early onset schizophrenia progresses into a severe chronic mental condition if left untreated within an appropriate timeframe [9]. As the early stages of schizophrenia decisively influence its course and prognosis [10], it is of vital importance to understand the features characterizing patients with early onset schizophrenia and establish a comprehensive nursing plan that considers their developmental stages and family system to provide family interventions [11].

Simulation is increasingly gaining a foothold in nursing education because of its advantages: it provides an alternative to clinical practice; it is a safe, efficient learning environment through repetitive experiences; and student training is often difficult to encounter in clinical settings [12]. Face-to-face simulation training programs using adult patients with schizophrenia have long been developed and utilized [13], and their effectiveness has also been demonstrated [14]. However, few programs involve patients with early onset schizophrenia and their families, and concerns have been raised about the ethical issues associated with using standard pediatric patients for simulation, with these programs requiring reliability and validity testing [15].

## 2. Background

To address this problem, simulation training can be established using avatars on a metaverse platform. “Metaverse” is a new virtual world and ecosystem driven by 3D technology, which creates a space that merges physical and virtual realities to allow users to interact with each other [16]. According to previous research exploring the applicability of a metaverse-based platform in education [17], a metaverse platform is configured to enable its users, represented by avatars, to engage in social interactions and emotional exchanges, and communicate with one another in a virtual world. In the nursing and medical fields, attempts have been made to develop programs using a new type of metaverse platform, but this undertaking is still in its early stages, with its learning effectiveness and efficacy yet to be validated [18]. Therefore, the need exists to develop a program using metaverse platforms, such as second life, Zepeto, gather town, Virbela, Jump virtual meetup, Icograms, spot, and ifland, in which avatars (representing platform users) can interact with one another to experience scenarios rarely encountered in clinical practice and difficult to implement in a face-to-face simulation and empirically verify its educational value.

Theory-based research has rarely been conducted to examine the metaverse-based simulation program’s operational components and strategies. Lindsey and Berger’s [19] experiential approach to instruction modeling recommends a three-step experiential instruction: framing the experience, activating the experience, and reflecting on the experience. Based on this model, this study incorporates Raskin and Rogers’ [6] person-centered therapy as an intervention strategy in designing a multi-access metaverse-based nursing simulation program for caring for children with early onset schizophrenia. Specifically, in this study, the core concepts of the person-centered therapy, sincerity, unconditional positive respect, and empathy were used in the counseling process of children with early onset schizophrenia and their guardians. According to the results of a systematic literature review on virtual reality simulation programs applied to nursing students and nurses [20], the theoretical foundations of the programs included in the analysis were insufficient, and that scenarios or cases are stored in the system and characterized as programs that interact with virtual objects. In this study, the person-centered therapy of Raskin and Rogers [6] was used as a theoretical basis for intervention, and differentiated from existing programs in that it develops a real-time multi-access program.

This study aims to develop a multi-access metaverse-based early onset schizophrenia nursing simulation program for nursing students to validate the effects of the transition to a virtual clinical experience as facilitated by the COVID-19 pandemic. Moreover, this work will verify the program’s effectiveness using the knowledge of patients with early onset schizophrenia (hereafter, “knowledge”), critical thinking ability, learning self-efficacy, communication ability, and learning satisfaction and confidence as outcome variables. Based on Jeffries’s [21] simulation framework and previous studies on schizophrenia nursing simulation programs, critical thinking ability, and the person-centered counseling theory was developed to identify knowledge about schizophrenia and enhancement of situational analysis competency. The effect of the applied interview was verified as communication ability, and the effects of enhancing competence in nursing patients with schizophrenia were learning self-efficacy, learning satisfaction, and confidence [13].

This study explored the following five specific hypotheses:

**H1.** 
*The experimental group receiving the multi-access metaverse-based early onset schizophrenia nursing simulation program will demonstrate a greater increase in its mean score for knowledge than the control group receiving only an online lecture.*


**H2.** 
*The experimental group will demonstrate a greater increase in the mean score for critical thinking ability than the control group.*


**H3.** 
*The experimental group will demonstrate a greater increase in the mean score for learning self-efficacy than the control group.*


**H4.** 
*The experimental group will demonstrate a greater increase in the mean score for communication ability than the control group.*


**H5.** 
*The experimental group will demonstrate a greater increase in the mean score for learning satisfaction and confidence than the control group.*


## 3. Materials and Methods

### 3.1. Study Design

This study developed a multi-access metaverse-based early onset schizophrenia nursing simulation program using a quasi-experimental design. Its effectiveness was tested using a nonequivalent control group pre-test/post-test design. Knowledge, critical thinking ability, learning self-efficacy, communication ability, and learning satisfaction and confidence were set as outcome variables (Figure 1).

### 3.2. Participants

The target population of this study was third-year nursing students who had just completed theoretical classes in schizophrenia and therapeutic communication-related mental health nursing. They were attending nursing schools offering Bachelor of Science in Nursing programs across South Korea. The inclusion criteria were nursing students who could use a smartphone or tablet PC. The exclusion criteria were those with experience in studying undergraduate courses other than nursing or working in clinical practice for one year or more, and those participating in a metaverse-related program or virtual reality game program, as such prior learning experiences were likely to affect the current results.

The number of participants was calculated with the G*power sample size calculator, version 3.1.9.4 [22]. For hypothesis testing, a mean-value comparison between the experimental and control groups was performed using independent *t*-tests as follows: (α) = 0.05, power (1-β) = 0.80, and median effect size (d) = 0.80. Consequently, the minimum required sample size per group was 26, totaling 52. The sample size per group was set at 29 (totaling 58). All 58 participants—29 per group and recruited online through a research notice uploaded to social media and websites used by nursing students—were included in the analysis.

### 3.3. Study Protocol

The program’s operational statistics were established based on Lindsey and Berger’s [19] experiential approach to instruction modeling implemented in three steps: framing, activating, and reflecting on the experience (Table 1). The scenarios for the multi-access metaverse-based early onset schizophrenia nursing simulation program were designed to be used on patients with early onset schizophrenia and their families based on Raskin and Rogers’ [6] (Table 2; Appendix A) person-centered therapy.

A six-member expert panel (one psychiatric nurse, one psychiatric specialist, one psychiatric nursing professor, one pediatric nursing professor, one professor with experience in metaverse program development, and one education science professor) assessed the drafted program scenarios’ validity by assigning an Item Content Validity Index (I-CVI) to each item; only the items assessed to reach the cut-off I-CVI value of 0.8 were included in the program’s final version. Participants were recruited using web addresses posted in a nursing students’ social media. Only those who consented to participate and left their personal contact information were enrolled. An online pre-test questionnaire was sent to all participants prior to the URL; it took approximately 20 to 30 min to complete the survey. At week 1 of the intervention, the online lecture (100 min) was provided to all participants of both experimental and control groups through the URL. At weeks 2 and 3, only the experimental group was provided with the simulation program debriefing (100 min; Table 2; Figure 2). For post-intervention feedback, the URL link to an online post-test questionnaire was sent to all participants one week after completing the intervention. After the post-test was completed for compensatory equilibration of treatment, a multi-access metaverse-based early-onset schizophrenia nursing simulation program used by the experimental group was provided to the control group.

### 3.4. Data Collection

Before initiating the research procedure, this study was approved by the appropriate Institutional Review Board (details blinded for peer review), which examined the study’s purpose, methodology, guarantee of the participants’ rights, and questionnaire. This study was a non-curricular activity, and participation was free from grades, while the autonomy in participation was guaranteed. The participants were informed of the guarantee of their rights and privacy protection, private data confidentiality, voluntary consent, and their right to withdraw their consent at any time without penalty. They were also informed that all data collected during the study would be used only for research purposes, with full anonymity and confidentiality ensured. This study conformed to the standards of the Declaration of Helsinki.

Participants’ knowledge, critical thinking ability, learning self-efficacy, communication ability, and learning satisfaction and confidence were measured in March and April 2022 using an online questionnaire by research assistants who had no conflict with the study subjects. The collected data were anonymized by research assistants immediately after data collection, and after the pre- and post-surveys were completed. The anonymized data were coded so that personal identifiable information could not be confirmed, thereafter the research data were managed.

### 3.5. Instruments

#### 3.5.1. Knowledge

This variable was measured using a tool developed by Seo and Kim [23] and based on the Assessment Tool for Schizophrenia [24] and a mental health handbook [25]. Subsequently, this tool was modified to suit this study’s purpose, and reliability and validity tests were conducted. The content validity of the modified tool was assessed by a psychiatric nursing professor and a psychiatrist. This knowledge measurement tool comprises 20 multiple-choice items (four response options) rated on a binary scale where zero and one indicate incorrect or correct answers, respectively. The total score ranges between 0 and 20; with a higher score indicating a higher knowledge of patients with early onset schizophrenia. The Kuder–Richardson-20 reliability coefficient was 0.71 at the time of the tool’s development and 0.91 in this study.

#### 3.5.2. Critical Thinking Ability

This variable was measured with the Critical Thinking Disposition Scale standardized by Yoon [26] through reliability and validity testing. This 27-item instrument comprises seven subscales: eagerness/curiosity (5 items), prudence (4 items), self-confidence (4 items), consistency (3 items), intellectual fairness (4 items), healthy skepticism (4 items), and objectivity (3 items). Each item is rated on a five-point Likert scale ranging from one (“strongly disagree”) to five (“strongly agree”), with total scores ranging from 27 to 135; a higher score indicates a higher critical thinking ability. Cronbach’s α was 0.84 in Yoon’s [26] study and 0.86 in this study.

#### 3.5.3. Learning Self-Efficacy

This was measured using the tool developed by Ayres [26] and adapted by Seo and Kim [23], who also verified its reliability and validity. Each item of this 10-item self-efficacy tool is rated on a seven-point Likert scale, ranging from one (“completely disagree”) to seven (“completely agree”), with the total score ranging between 10 and 70; a higher score indicates a higher learning self-efficacy. Cronbach’s α was 0.94 at the time of its development [27] and 0.95 in this study.

#### 3.5.4. Communication Ability

This was measured using the Communication Ability Scale developed by Lee [28] to assess nurses’ facilitative communication ability by adapting the Index of Communication developed by Carkhuff [29] and standardizing it through reliability and validity testing. This 35-item scale comprises seven subscales: empathic understanding (5 items), positive regard (5 items), authenticity (5 items), concreteness (5 items), confrontation (5 items), self-disclosure (5 items), and immediacy (5 items). Each item has five response options and is rated by assigning a numeric level (1 = level 1, 5 = level 5). The total scores ranged from 35 to 175 points; the higher the total score, the higher the communication ability. Cronbach’s α was 0.75 at the time of development and 0.70 in this study.

#### 3.5.5. Learning Satisfaction and Confidence

This was measured using the Korean version [30] of the Student Satisfaction and Self-Confidence in Learning Scale developed by the National League for Nursing [31] to measure nursing students’ learning satisfaction with and confidence in simulation-based teaching methods. This 13-item instrument includes two subscales: satisfaction with learning (5 items) and self-confidence in learning (8 items). Each item is rated on a five-point Likert scale, ranging from one (“strongly disagree”) to five (“strongly agree”), with total scores ranging from 13 to 65 points; a higher score indicates a higher level of satisfaction with and self-confidence in simulation as a teaching modality. Cronbach’s α coefficients ranged from 0.87 to 0.94 and 0.72 to 0.89 in the studies using the original and adapted versions, respectively, and 0.93 in this study.

### 3.6. Data Analysis

The data collected were analyzed with the statistical package PASW 27.0 for Windows (SPSS Inc., Chicago, IL, USA). As participants were normally distributed, a parametric analysis was performed. Chi-squared, Fisher’s exact, and independent *t*-tests were used to test the homogeneity between the experimental and control groups regarding general characteristics and dependent variables (knowledge, critical thinking ability, learning self-efficacy, communication ability, and learning satisfaction and confidence). Independent *t*-tests and analyses of covariance were then conducted to analyze within-group differences from pre- to post-test scores and the between-group differences in the mean pre-test/post-test score differences in the dependent variables.

## 4. Results

### 4.1. Participants’ General Characteristics and Homogeneity Testing

Participants were 58 third-year nursing students (29 each in the experimental and control groups). Homogeneity between the two groups was confirmed in homogeneity testing, in which no significant differences were found in their general characteristics: sex (*p =* 0.470), satisfaction with nursing major (χ^2^(2, *N* = 58) = 3.98, *p =* 0.264), satisfaction with clinical practice (χ^2^(2, *n* = 58) = 2.98, *p =* 0.394), demand for metaverse education (*p =* 0.237), and demand for EduTech (*p =* 0.491; Table 3). Regarding the pre-test dependent variables, homogeneity between the two groups was established in critical thinking ability (*t*(56) = 0.94, *p =* 0.354), learning self-efficacy (*t*(56) = 0.57, *p =* 0.570), communication ability (*t*(56) = 0.23, *p =* 0.823), and learning satisfaction and confidence (*t*(56) = 0.35, *p =* 0.711), with no significant differences found between the two groups. However, a significant intergroup difference was found in the knowledge variable (*t*(56) = 2.15, *p =* 0.036), indicating that homogeneity was not demonstrated (Table 3).

### 4.2. Verifying the Experimental and Control Groups’ Effectiveness

Table 4 outlines this statistical analysis results. No significant differences were found between the experimental and control groups in those between the pre- and post-test mean scores of learning self-efficacy (*t*(56) = 0.24, *p =* 0.814) and learning satisfaction and confidence (*t*(56) = 1.37, *p =* 0.682). However, significant intergroup differences were found in those of knowledge (*t*(56) = 2.13, *p =* 0.038), critical thinking ability (*t*(56) = 1.29, *p =* 0.013), and communication ability (*t*(56) = 1.10, *p =* 0.045). More specifically, the exogenous variables were treated as covariates in the ANCOVA analysis, which found the experimental group had significantly increased knowledge (*F*(2) = 1.15, *p =* 0.046), critical thinking ability (*F*(2) = 9.86, *p =* 0.038), and communication ability (*F*(2) = 0.05, *p =* 0.043; Table 4) compared to that of the control group.

## 5. Discussion

In the psychiatric department, nurses have the specificity of self-therapeutic use as a primary therapeutic tool, surpassing the role of the nursing provider [32]. Therapeutic communication between nurses and patients in mental health management is at the core of positive patient outcomes [33]. However, the COVID-19 pandemic has caused an unprecedented shift to online practice, with limited access to traditional clinical practice and face-to-face simulations. Thus, it is necessary for nursing students not prepared for psychiatric patients to train in therapeutic communication to avoid any harm while communicating. Accordingly, to promote perceived real experiences, a virtual simulation based on a digital learning environment using various devices—computers, tablets, screens, and head-mounted displays—was introduced for use in psychiatric nursing practice [34].

Consequently, in this study, the experimental group—which received the multi-access metaverse-based early onset schizophrenia nursing simulation program using the experiential approach to instruction model [8]—significantly improved their knowledge scores compared with the control group, which received only an online lecture. This parallels the findings from a previous study [35] and meta-analysis [36], which found that undergraduate nursing students significantly increased their knowledge scores after receiving web-based experiential learning and virtual reality simulation programs. This result can be attributed to students’ interactions and experience with applying their lecture-based theoretical knowledge to a virtual situation with multiple simultaneous users, represented by avatars. Kyaw et al. [37] indicated that a highly interactive virtual reality simulation further increases the efficiency of knowledge. Similarly, this study’s metaverse program participants stated in the post-intervention feedback survey that in addition to theoretical instruction, they owed a great part of their learning to patient interviews through virtual simulation. An immediate debriefing process was required after the virtual simulation, and through debriefing, learners’ knowledge gaps were identified and learning was improved [38]. It seemed necessary for debriefers to prepare standardized and structured questions so that self-awareness and reflection could be promoted by providing debriefing in a timely manner immediately after the virtual simulation [39]. However, in the pre-test, there was a significant difference between the experimental group and the control group; therefore, homogeneity was not secured. ANCOVA statistical analysis was performed to correct for this; however, interpretation of the results is necessary.

A systematic review [20] also focused on the advantages of using virtual worlds in nursing education, and it highlighted their effectiveness in improving cognitive function. This suggests that virtual spaces can potentially be applied to experiential learning in nursing training. Moreover, this study revealed that improved knowledge in the experimental group may be attributed to the use of an evaluation tool for the nursing care of patients with early onset schizophrenia—provided to assess patients’ conditions—and a debriefing process to reflect on the mental health nursing interventions provided to the avatars of schizophrenia patients in the metaverse. The reflection-based learning approach helped link experience and knowledge [40]. In the future, metaverse program designs should consider the debriefing standards and methods of the International Nursing Association of Clinical and Simulation Learning’s (INACSL) best practices [41] in considering the learning target and training topic.

The experimental group also significantly improved in their critical thinking ability compared with the control group. In previous research, case-based learning was more efficient in improving critical thinking ability than traditional lecture-based instruction [42], and active and reflective learning was mentioned as an important learning modality for improving critical thinking ability [43]. In this study’s virtual simulation program, a psychiatric nurse presented realistic scenarios to the nurse avatars in the multi-access metaverse space, helping develop their care processes and decisions to maximize therapeutic effects in virtual clinical situations. In encouraging their deeper self-reflection by posing concrete reflective questions, this structured debriefing approach significantly improved their critical thinking ability. The INACSL Standards Committee [41] has highlighted the need for structured, well-designed debriefing and reflection processes [33], which were considered in this study’s learning design to enable the application of structured and systemized “gather, analyze, summarize” debriefing in virtual reality.

While this significantly improved the experimental group’s mean critical thinking score, this group demonstrated no more significant improvement in learning self-efficacy than the control group. In a previous study of virtual simulation targeting nursing students [44], self-efficacy was improved, which differed from the current results. Mabry et al. [44] applied a framework that combined Ericsson’s [45] theory of planned practice and Bandura et al.’s [46] self-efficacy theory to facilitate learners to achieve high-level mastery through a deliberate practice process. However, this study’s participants were junior students with no experience in clinical practice, and it was difficult for them to attempt therapeutic communication with patients without orientation or experience in the psychiatric department. A conceptual understanding was achieved through online lecture, but their immediate use of such knowledge was limited. In Bandura et al.’s [46] self-efficacy theory, indirect experience and an active achievement process improves self-efficacy. A previous study [47] indicated that it is possible to experience a sense of accomplishment and increase self-efficacy through repeated simulation experiences. However, the participants received simulation training only once and performed only two virtual simulation training sessions. This alone is insufficient for a successful experience and does not improve self-efficacy. Future research should include a process of providing indirect experience showing therapeutic communication between actual nurses and patients and enabling sufficient practice among learners. Additionally, a step-by-step approach and repeat learning opportunities should be provided so learners can have a successful experience by using strategies to enhance self-efficacy.

The experimental group also outperformed the control group in its rate of significant improvement in communication abilities. In previous studies of interactive simulation training for healthcare professionals [48], role-play and interaction simulations identified potential individual weaknesses and improved authentic communication skills, thus supporting our results. In this study, we developed situational scenarios to provide the experimental group with opportunities to experience sincerity, unconditional positive regard, and empathy—the core conditions of person-centered therapy—and share their experiences through exchanges in the virtual simulation training’s debriefing process. It seemed that having the study subject as a counselor who applied the person-centered counseling theory that provided patient-centered supportive communication reduced the defensive attitude of the client and improved effective communication. A previous study [49] also confirmed that patient-centered communication in mental health resulted in positive communication, even for children and adolescents. Specifically in this study, all learners were encouraged to use therapeutic communication skills (listening, open-ended questions, restatement, clarification, reflection, focus, awareness sharing, topic identification, silence, humor, informational, suggestion, or confrontation, among others) and discuss the appropriateness of the techniques used throughout the debriefing process.

Kalisch et al. [50] noted that team training in a virtual world is an innovative learning method that allows learning without spatiotemporal constraints through participation as virtual characters; the authors emphasized that it provides a safe environment with a greater capacity for multiple access users. Activities in a virtual metaverse environment, such as therapeutic communication between the patient (child with schizophrenia) and learner (nursing student) avatars, enable the latter to freely interact with one another, building a support system among small groups. This environment can reduce the frustration associated with failure and mistakes while encouraging communication. In the post-intervention feedback survey, the participants opined that it was possible to conduct interviews with less tension because the avatars appeared friendly, and no actual contact was involved in the virtual world. A previous study [34] observed that virtual simulation allows learners to immerse themselves in training with psychological stability because they can receive training in a safe environment where mistakes are allowed without worrying about errors in front of fellow students or trainers. However, as this study did not provide an opportunity for patient-to-nurse one-to-one communication similar to an actual clinical situation, an additional opportunity for in-depth therapeutic communication should be provided in the future.

Regarding the “learning satisfaction and confidence” variable, no significant improvement was found in the experimental group, which received the multi-access metaverse-based early onset schizophrenia nursing simulation program, compared with the control group, which received only an online lecture on patients with schizophrenia. In a meta-analysis [36] of virtual reality nursing education programs, no significant difference was observed in learning satisfaction and confidence compared to the control group, supporting our results. However, a previous study [51] noted that using an immersive virtual environment significantly improved learning satisfaction compared to the existing education method. In a systematic literature review [52], the more immersive the methods used, the more positive the learning experiences, which improved satisfaction. Our results may be attributable to the multi-access design, in which multiple learners can access the program simultaneously and communicate therapeutically with the patient avatar; this can cause issues with “howling” and sound interruption, and technological limitations exist in implementing avatars’ expressions and movements. Participants in the post-intervention feedback survey mentioned difficulties in empathizing with the patient because they could not see the patient’s expressions and actions. A previous study [53] also reported that technological limitations in implementing a metaverse made it difficult to discern facial cues and nonverbal actions and operate a natural interface, leading to user dissatisfaction. Therefore, it is necessary to further develop programs based on upgraded technology that enables avatars to change their facial appearance and show personalized expressions and actions [53].

Participants’ immersion in the simulated situation was adversely affected by a cognitive gap owing to the unfamiliar manipulation of the left and right controllers to move the avatar. Students’ immersion and sense of presence in learning in a non-face-to-face learning environment are important factors in their learning satisfaction [52]. Therefore, future applications of metaverse-based simulation training will have to be preceded by a stable network and sufficient practice to allow the user to become accustomed to the metaverse platform and program operation. Furthermore, this study confirmed the need to select and utilize a nursing training topic unhindered by a metaverse’s technical limitations, or consider the application of a hybrid simulation method to increase the learner’s sense of immersion and satisfaction. An integrative review [54] found that repeated simulation experiences improve confidence. In this study, the students had no clinical practice experience, and no simulation training (face-to-face or non-face-to-face) was performed. Moreover, the simulation training was performed once for each scenario, with different scenarios. It is necessary to design the program to increase the number of repeated experiences and similar cases for a successful outcome. This study incorporated the Zepeto^®^ World metaverse platform, which can be used free of charge by downloading the application to a smartphone. Other Big Tech companies are also rushing to develop various metaverse platforms, which could prove useful. Accordingly, it is necessary to prepare for any future pandemics by selecting a platform after considering its purpose and learner friendliness and developing a metaverse program based on virtual experiences from various cases.

Considering the five specific hypotheses explored in this study, the metaverse-based simulation program in its current mode can improve the knowledge, critical thinking, and communication skills of psychiatric nursing students. However, some modifications may be needed before it can facilitate learning satisfaction, confidence, and self-efficacy.

## 6. Conclusions

Considering COVID-19 and the subsequent paradigm shift of education to IT-based e-learning, nursing students’ clinical education in a psychiatric ward has been limited or discontinued, precipitating the need to develop EduTech content capable of replacing clinical practice.

In this study, we developed a multi-access, metaverse-based early onset schizophrenia nursing simulation program based on Raskin and Rogers’ [6] person-centered therapy to improve nursing students’ therapeutic communication abilities when caring for psychiatric patients. We derived some positive effects from implementing the simulation program that indicate that this program can be used as an alternative option in high-quality nursing education without spatiotemporal constraints to replace clinical psychiatric practice. However, owing to technological limitations, the avatars’ facial expressions and bodily postures were limited and are far from natural compared with that of a standard patient. To best utilize information and communication technology (ICT) in nursing education, content developers and operators should enhance software utilization capabilities and identify and prepare for problems occurring in the program’s operational process before its use in education. This problem must be addressed by systematic training and sufficient testing to improve the users’ understanding of ICT. Additionally, as a clinical alternative training plan for nursing students in preparation for a future pandemic, it is necessary to develop various metaverse-based simulation training programs; identify their utility, feasibility, and effectiveness; and augment and refine them.

This study had some potential limitations that can impact future research directions. First, ANCOVA statistical analysis was performed to compensate for the lack of homogeneity between the experimental group and the control group in knowledge measured by the dependent variable; nevertheless, care must be taken in interpreting the results. Second, our results may not be generalizable, as the sampled control and experimental groups are not representative of all eligible nursing students in South Korea. Third, pre- and post-testing program effectiveness could not confirm the sustainability of its effects without follow-up test data. Hence, a subsequent study is necessary to evaluate the long-term effects of this intervention program and its clinical efficacy. Fourth, in the study, providing only lectures to the control group and comparing effectiveness with the experimental group has limitations in making sufficient comparisons. Moreover, the experimental and control groups were exposed to different online and offline environments and different feedback and debriefing modalities. Future researchers must examine the intergroup differences in effectiveness after providing participants in both groups with practical training on the same subject and metaverse, lab, and clinical settings.

## Figures and Tables

**Figure 1 ijerph-20-00449-f001:**
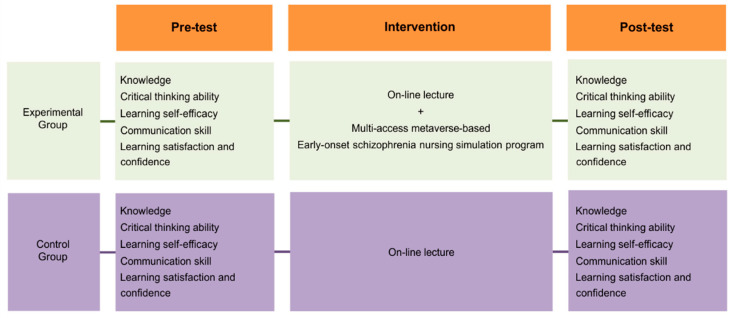
Research design. The online lecture includes the following topics: characteristics and major symptoms of patients with early onset schizophrenia (25 min), treatment and intervention methods for patients with early onset schizophrenia (25 min), therapeutic communication strategies for patients with early onset schizophrenia and their caregivers (listening, open-ended questions, restatement, clarification, reflection, focusing, sharing perceptions, identifying topics, silence, humor, the provision of information, suggestions, and confrontation; 50 min) through the online Learning Management System.

**Figure 2 ijerph-20-00449-f002:**
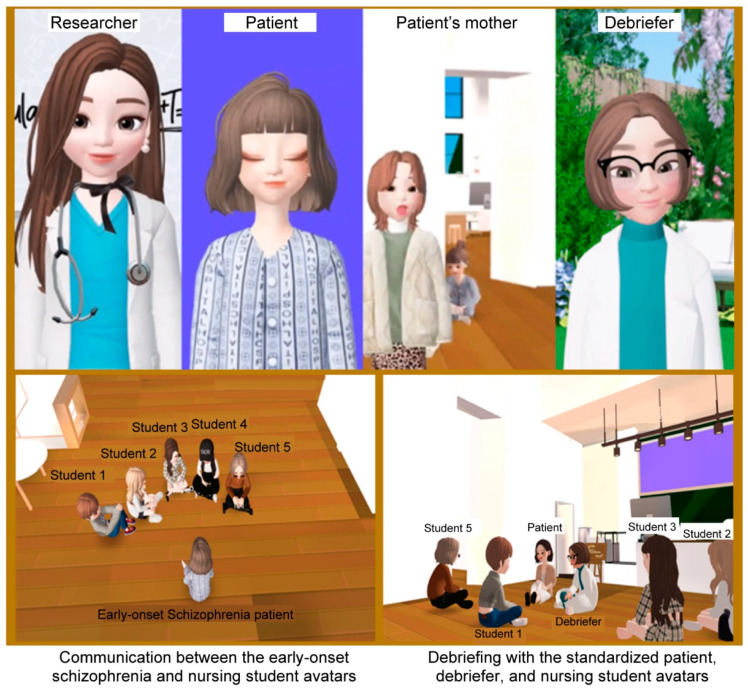
Screenshot of the early onset schizophrenia nursing simulation program in the metaverse (Zepeto^®^ World).

**Table 1 ijerph-20-00449-t001:** Multi-access metaverse-based early onset schizophrenia nursing simulation program using Lindsey and Berger’s [18] experiential approach to an instructional model.

Framing the Experience	Activating the Experience	Reflecting on the Experience
▪ Objectives Fostering appropriate therapeutic communication to establish a rapport with the patient Fostering empathy, acceptance, support, and encouragement, without criticism and judgment, to identify patients’ mental health issues	▪ Authenticity Constructing a closed pediatric ward in the psychiatric department within the metaverse (the Naver Inc. Zepeto^®^ World)	▪ Guiding as facilitator Configuring the facilitator avatar to lead the learner avatars to reflect on their activities conducted in the metaverse
▪ Assessment criteria Utilizing nursing skill evaluation criteria for patients with schizophrenia (hallucination management, suicide risk and safety management, delusion management, violence risk and safety management, medication management, communication abilities, and attitudes)	▪ Required decisions Cognition/knowledge: identifying the mental health issues of an early onset schizophrenia patient; helping the patient recognize the problem and educate them Application/technology: identifying the mental health issues of an early onset schizophrenia patient and making decisions given the modality of therapeutic communication	▪ Providing a reflective question What are the elements required to address schizophrenia in the metaverse?
▪ Defined social expectations Psychiatric ward nurse capable of therapeutic communication	▪ Be problem-oriented Capturing patients’ abnormal behavioral characteristics through therapeutic communication techniques Providing patients with mental health nursing interventions through therapeutic communication	▪ Providing a supportive community Encouraging a support system in which the students form a small group to solve the encountered problems together, and support and encourage one another
	▪ Optimal difficulty Setting the difficulty level as one that can be solved by nursing students after attending an online lecture on therapeutic communication theory and nursing care for patients with schizophrenia	

**Table 2 ijerph-20-00449-t002:** Contents of a multi-access metaverse-based early onset schizophrenia nursing simulation program based on Raskin and Rogers’ [6] person-centered counseling theory.

Week	Stage	Intervention Content	Elements	Time
1	Pre-briefing	Orientation - Explaining the goal and program’s objectives - Demonstrating how to use Zepeto^®^ World - Small-group activities: assigning users to six groups (four or five per group) and allocating roles	Naver Inc.’s Zepeto^®^ World (hospital and conference rooms 1–6) Google Docs	50 min
		Online lecture - Characteristics and major symptoms of patients with early onset schizophrenia - Treatment and intervention methods for patients with early onset schizophrenia - Therapeutic communication based on person-centered counseling theory - Therapeutic communication strategies for patients with early onset schizophrenia and their caregivers	Online Learning Management System	100 min
2	Simulation running	Scenario 1: An initial interview with a 17-year-old girl on the day of her admission to a pediatric ward in the psychiatric department with a diagnosis of early onset schizophrenia for symptoms including hallucinations and delusions that began six months before the diagnosis - Participants and roles: standardized patient (SP) avatar 1 (patient with early onset schizophrenia with psychotic symptoms in an initial hospitalization), SP avatar 2 (mother of the patient with early onset schizophrenia), and student nurse avatars 3 to 7 (psychiatric nurses) - Student nurse 1: Obtains the patient’s general information, including past history of hospitalization; explains the closed ward; and provides therapeutic communication related to the symptoms’ time of occurrence - Student nurse 2: Assesses the patient’s mental health status and provides therapeutic communication to assess mood conditions, auditory hallucinations, and delusions - Student nurse 3: Assesses the patient’s mental health status and provides therapeutic communication to determine if any suicidal ideation exists - Student nurse 4: Assesses the patient’s mental health status and provides therapeutic communication related to cognition (memory) - Student nurse 5: Assesses the patient’s mental health status and provides therapeutic communication related to judgment, insight, and medication education Scenario 2: An initial interview with the mother of the patient with early onset schizophrenia on the day of hospitalization - Participants and roles: SP avatar 2 (mother of the patient with early onset schizophrenia with psychotic symptoms, paranoia, and hallucinations) and four to five student nurse avatars - Student nurse 1: Provides initial inpatient support for the caregiver and caregiver education, and a therapeutic conversation regarding the patient’s abnormal behaviors - Student nurse 2: Assesses the patient’s growth and development history and provides relevant therapeutic communication - Student nurse 3: Assesses the patient’s family history and provides relevant therapeutic communication - Student nurse 4: Provides disease-related information, family support, and relevant therapeutic communication - Student nurse 5: Explains the treatment plan in the closed ward and provides relevant therapeutic communication	Naver Inc.’s Zepeto^®^ World (hospital)	50 min
	Debriefing	Small-group reflective activities: A “GAS” debriefing with the debriefer, standardized patient, and students ▪ Gather What was your metaverse simulation’s outcome? What are your feelings about the simulation you have just completed? Let us discuss why you had these feelings about the simulation scenario. ▪ Analyze What was the virtual simulation about? What was the condition of the patient with early onset schizophrenia in the virtual simulation? What were the nursing issues to be resolved?How did you feel when you were communicating with the patient and caregiver? Did you provide therapeutic communication? What do other team members think about that statement? What difficulties did you encounter during the metaverse simulation? Please describe in detail. ▪ Summarize Which intervention was provided to the patient with early onset schizophrenia and the caregiver in the virtual simulation today, and on what grounds did you provide that intervention? Which intervention did you fail to provide to the patient with early onset schizophrenia and the caregiver in the virtual simulation today? Please explain why you could not provide that intervention. What do you think were well-resolved features of this virtual simulation? Let us discuss the individual key take-home points we learned through this virtual simulation with the early onset schizophrenia patient’s avatar. What points should be considered to overcome the shortcomings found in today’s simulation scenario in the future? Each student should then provide a self-evaluation of their achievement level given the program learning goal.	Naver Inc.’s Zepeto^®^ World (conference rooms 1–6) Google docs	100 min

**Table 3 ijerph-20-00449-t003:** Homogeneity test of general characteristics of participants and dependent variables (*N* = 58).

Characteristics	Categories	Experimental Group (*n* = 29)	Control Group (*n* = 29)	χ^2^ or *t*	*p*
		*n* (%)	*n* (%)		
Sex	Male	6 (20.7)	3 (10.3)		0.470
	Female	23 (79.3)	26 (89.7)		
Satisfaction with nursing major	Dissatisfied	0 (0.0)	1 (3.4)	3.98	0.264
	Neutral	6 (20.7)	4 (13.8)		
	Satisfied	23 (79.3)	24 (82.8)		
Satisfaction with clinical practice	Dissatisfied	2 (6.9)	1 (3.5)	2.98	0.394
	Neutral	12 (41.4)	15 (51.7)		
	Satisfied	15 (51.7)	13 (44.8)		
Demand for EduTech (e.g., a metaverse)	Yes	29 (100.0)	26 (89.7)		0.237
	No	0 (0.0)	3 (10.3)		
Metaverse education helps clinical nursing practice	Yes	29 (100.0)	27 (93.1)		0.491
	No	0 (0.0)	2 (6.9)		
Knowledge, *M* (*SD*)		11.86 (2.08)	13.00 (1.95)	2.15	0.036 *
Critical thinking ability, *M* (*SD*)		101.14 (9.78)	98.55 (11.23)	0.94	0.354
Learning self-efficacy, *M* (*SD*)		58.83 (6.29)	57.69 (8.67)	0.57	0.570
Communication ability, *M* (*SD*)		132.79 (8.82)	133.28 (7.47)	0.23	0.823
Learning satisfaction and confidence, *M* (*SD*)		53.65 (7.14)	53.18 (5.69)	0.35	0.711

* In the preliminary homogeneity test of the experimental group and the control group, the knowledge variable showed a significant difference between the two groups; therefore, homogeneity was not secured.

**Table 4 ijerph-20-00449-t004:** Comparison of dependent variables among experimental and control groups (*n* = 58).

Variables	Pre-Test			Post-Test			Mean Change			ANCOVA
	EG (*n* = 29) *M* (*SD*)	CG (*n* = 29) *M* (*SD*)	*t* (*p*)	EG (*n* = 29) *M* (*SD*)	CG (*n* = 29) *M* (*SD*)	*t* (*p*)	EG (*n* = 29) *M* (*SD*)	CG (*n* = 29) *M* (*SD*)	*t* (*p*)	*F* (*p*)
Knowledge	11.86 (2.08)	13.00 (1.95)	−2.15 (0.036)	14.38 (3.09)	13.72 (2.59)	0.88 (0.385)	2.52 (3.52)	0.72 (2.85)	2.13 (0.038)	1.15 (0.046)
Critical thinking ability	99.14 (9.78)	98.55 (11.23)	0.94 (0.654)	105.48 (10.57)	99.00 (11.10)	2.28 (0.027)	6.34 (10.79)	0.45 (12.24)	1.29 (0.013)	9.86 (0.038)
Learning self-efficacy	58.83 (6.29)	57.69 (8.67)	0.57 (0.570)	59.66 (8.31)	58.10 (7.78)	0.73 (0.466)	0.83 (6.71)	0.41 (6.65)	0.24 (0.814)	3.37 (0.072)
Communication ability	132.79 (8.82)	133.28 (7.47)	−0.23 (0.823)	136.72 (10.41)	133.48 (7.74)	0.32 (0.041)	3.93 (11.73)	0.21 (8.57)	1.10 (0.045)	0.05 (0.043)
Learning satisfaction and confidence	53.65 (7.14)	53.18 (5.69)	0.35 (0.711)	54.86 (6.60)	53.69 (6.81)	0.67 (0.508)	1.21 (9.45)	0.51 (9.18)	1.37 (0.682)	1.51 (0.577)

## Data Availability

The data that support the findings of this study are available from the corresponding author upon reasonable request.

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
