# Peer review of "Efficacy Testing of a Multi-Access Metaverse-Based Early Onset Schizophrenia Nursing Simulation Program: A Quasi-Experimental Study"

_ijerph, 2022, doi:10.3390/ijerph20010449_

Round 1

Reviewer 1 Report

The study is well written, concise. The methodology is clear and the materials provided would help guide someone to replicating this simulation for their own cohort

My primary concern is an ethical one. The IRB approval notwithstanding, generally in education research, BOTH the control and the intervention group would be afforded the opportunity to participate in the intervention. For instance, after the study intervention ends, those in the control group should be exposed to the intervention, regardless of whether planned data collection was to be done or not. Without that, there is an inequity of the learning experience in the education programs.

My other concern is the fact the intervention group had a significantly higher Knowledge prescore than the control group. How might this impact the results?

The only other critique I have focuses on some minor edits that should be considered. See below

Page

Line

Comment

1

36-38

It’s interesting that South Korea is not allowing students back into the psychiatric setting due to the vulnerability of patients.

2

68

That statement regarding the “effectiveness” of F2F sim training programs vis a vis schizophrenia is lacking a reference or two.

2

75-77

Is it possible to give examples of a metaverse, like Second Life for instance, or others

5

Tables 1 & 2

Suggest using a left alignment rather than center for this table’s content.

Table 4

It looks like there was a significant difference in the baseline knowledge of students in the control and experimental groups, is that correct? This should be commented on.

PRETEST EG 11.86 vs 13. P=0.036    How did that happen? Could anxiety of the upcoming sim participation have played a role

5

128-131

Some of the writing is very passive and could be strengthened with an active voice, as in: “Accordingly, to promote perceived real experiences, a virtual simulation based on a digital learning environment using various devices….   has been introduced for use in psychiatric nursing  practice [32].”   

Suggest use of WAS instead of HAS BEEN

 Consider whether the following references are needed. They seem rather dated.

Ayres, H.W. 2005. Factor related to motivation to learn and motivation to transfer learning in a nursing population (Publication 369 No. 3162411) [Dissertation, North Carolina State University]. ProQuest Dissertations Publishing. https://reposi- 370 tory.lib.ncsu.edu/handle/1840.16/3773 371 26. Lee, J.J. A Study on Nurses Promoting Communication and Personal Relations [Master’s Thesis]; Jeju National University, 2005.

Author Response

The responses are provided in tabular form, in the attached file.

Reviewer 2 Report

The topic of this manuscript is timely and meaningful because virtual simulation education is growing very fast due to COVID-19 situation. However, there are some areas to be clarified and supplemented to make this manuscript stronger.

Below are points that need to be clarified and addressed.

-         In the introduction and background, there is a need for an explanation of why child population and why schizophrenia patient were included in this study. Also more explanation is needed why the variables and instruments in this study were chosen.

-         For the communication ability variable, the instrument for ‘facilitating communication ability’ were used. Is it different from communication skill? What is ‘facilitating’ mean in this instrument?

-         Control group only had lectures. Is it enough as a control group?

-         Authors mentioned person-centered therapy several times, but the definition and the core concept were not fully explained. Also what aspects of person-centered therapy were applied in this study?

-         There were many metaverse simulations seen in nursing education these days. What are the differences of this study or this simulation from the previous simulation educations?

-         In the discussion section, more in-depth analysis and discussion is required to fully explain the efficacy of simulation education that was developed in this study.

-         In some parts, a clear description of the result is needed. For example, “experimental group exhibited greater significant differences between the…” please rewrite this sentence.

-         The intervention was applied to students, but ethical consideration about ‘coersion’  was not mentioned. Please explain whether the students were free from the grade and researchers were not closely involved in data collection and evaluation.

Thank you for the opportunity to review the manuscript.

Author Response

(The authors gave the same response as above.)
